# Clonal Spread of Carbapenem-Resistant *Klebsiella pneumoniae* Sequence Type 11 in Chinese Pediatric Patients

Xiong Liu,[a] Kaiying Wang,[a] Jiali Chen,[c] Jingwen Lyu,[b] Jinhui Li,[a] Qichao Chen,[a] Yanfeng Lin,[a] Benshun Tian,[b] Hongbin Song,[a] Peng Li,[a] Bing Gu[b]

ᵃChinese PLA Center for Disease Control and Prevention, Beijing, China

ᵇLaboratory Medicine, Guangdong Provincial People's Hospital, Guangdong Academy of Medical Sciences, Guangzhou, China

ᶜChina Medical University, Shenyang, China

Xiong Liu, Kaiying Wang, and Jiali Chen contributed equally to this work. Author order was determined both alphabetically and in order of increasing seniority.

**ABSTRACT** *Klebsiella pneumoniae* often causes life-threatening infections in patients globally. Despite its notability, little is known about potential nosocomial outbreak and spread of *K. pneumoniae* among pediatric patients in low- and middle-income countries. Ninety-eight *K. pneumoniae* strains isolated from pediatric patients in a large general hospital in China between February 2018 and May 2019 were subjected to nanopore and Illumina sequencing and genomic analysis to elucidate transmission and genetic diversity. The temporal distribution patterns of *K. pneumoniae* revealed a cluster of sequence type 11 (ST11) strains comprising two clades. Most inferred transmissions were of clade 1, which could be traced to a common ancestor dating to mid-2017. An infant in the coronary care unit played a central role, potentially seeding transmission clusters in other wards. Major genomic changes during the outbreak included chromosomal mutations associated with virulence and gains and losses of plasmids encoding resistance. In summary, we report a nosocomial outbreak among pediatric patients caused by clonal dissemination of KPC-2-producing ST11 *K. pneumoniae*. Our findings highlight the value of whole-genome sequencing during outbreak investigations and illustrate that transmission chains can be identified during hospital stays.

**IMPORTANCE** We report a nosocomial outbreak among pediatric patients caused by clonal dissemination of *bla*KPC-2-carrying ST11 *K. pneumoniae*. Strains of various sequence types coexist in the complex hospital environment; the quick emergence and spread of ST11 strains were mainly due to the plasmid-mediated acquisition of resistance genes. The spread of hospital infection was highly associated with several specific wards, suggesting the importance of genomic surveillance on wards at high risk of infection.

**KEYWORDS** antimicrobial resistance, hospital outbreak, *Klebsiella pneumoniae*, whole-genome sequencing, transmission

**K**lebsiella pneumoniae is a common opportunistic bacterial pathogen that causes a wide range of diseases, including pulmonary (1), urinary tract (2), and bloodstream (3) infections and sepsis (4). Over the past decade, *K. pneumoniae* has become a major clinical and global health concern due to the rapid emergence and worldwide spread of strains resistant to carbapenem antibiotics (5, 6). Carbapenems are antimicrobials with proven efficacy in serious infections caused by extended-spectrum β-lactamase (ESBL)-producing bacteria. They possess broad-spectrum antibacterial activity which confers protection against most β-lactamases, such as metallo-β-lactamase (MBL) as well as extended spectrum β-lactamases. Therefore, carbapenems are used as the last-

Address correspondence to Hongbin Song, hongbinsong@263.net, Peng Li, jiekenlee@126.com, or Bing Gu, gb20031129@163.com.

The authors declare no conflict of interest.

resort antibiotics for treating bacterial infections. The World Health Organization listed ESBL-producing and carbapenem-resistant *K. pneumoniae* (CRKP) as a critical public health threat.

Infections caused by CRKP strains often have few treatment options and are associated with high mortality rates (7). Nosocomial outbreaks of CRKP infections have been reported in many countries, including the United States (8, 9), Poland (10), and Vietnam (11, 12). Carbapenem resistance in *K. pneumoniae* isolates increased in China from 2.5% to 15.8% between 2008 and 2018 (13). Sequence type 11 (ST11) is the most prevalent CRKP ST in China, and $bla_{KPC-2}$ is one of the most common drug resistance genes (14, 15).

Patients in neonatal intensive care units are at high risk of nosocomial CRKP infections due to an immature immune system and frequent invasive procedures (16, 17). Despite a significant reduction over the past 2 decades (18), the global burden of infant mortality remains significant, with a rate of 28 deaths per 1,000 live births in 2019 (19). Fatalities are caused primarily by multidrug-resistant strains (20). Most recorded deaths occur in low- and middle-income countries (21). The rate of carbapenem resistance in *K. pneumoniae* isolates from Chinese pediatric patients increased dramatically, from 2.2% to 25.4%, between 2005 and 2017 (22). The control of nosocomial infections among pediatric patients has become an urgent health challenge that must be addressed.

The development of effective strategies to control hospital outbreaks requires a full understanding of how nosocomial pathogens spread between patients. Whole-genome sequencing has been widely used in tracking hospital outbreaks of CRKP (23, 24). Long-read sequencing techniques such as Oxford Nanopore sequencing can be highly useful in resolving and tracking the dynamics of plasmids during transmission (25). Studies of CRKP hospital outbreaks have focused primarily on adult patients (26, 27). However, higher morbidity and mortality have been observed among CRKP-infected pediatric patients (28). The current knowledge regarding the transmission characteristics of this pathogen among pediatric patients is limited.

In this study, we have performed short-read sequencing (Illumina) and long-read sequencing (MinION; Oxford Nanopore Technologies) to investigate a nosocomial outbreak and clonal spread of ST11 CRKP among pediatric patients. An infant in the coronary care unit (CCU), who was found to play a central role early in the outbreak, may have seeded transmission clusters in other wards.

## RESULTS

**General features of patients and strains.** Ninety-eight *K. pneumoniae* strains (see Table S1 in the supplemental material) were isolated from 86 hospitalized children, 8 and 2 of whom had two and three strains, respectively. The median age was 4 months, ranging from 2 days to 13 years. The vast majority of strains (86.7% [85/98]) were isolated from children under 1 year of age (Fig. 1A), 9 from children between 2 and 6 years of age, and 4 from children aged between 12 and 13 years. The male-to-female ratio was 1.6. The increase in the number of *K. pneumoniae* strains was observed since June 2018 and reached its peak at January 2019 (Fig. 1B); a clonal spread *K. pneumoniae* strains was suspected.

*K. pneumoniae* strains (Table S1) were collected from sputum (72.4% [71/98]), blood (9.2% [9/98]), urine (8.2% [8/98]), pus (3.1% [3/98]), ascites (2.0% [2/98]), secretions (2.0% [2/98]), bronchoalveolar lavage fluid (1.0% [1/98]), cerebrospinal fluid (1.0% [1/98]), and endotracheal tube aspirate (1.0% [1/98]). The strains were isolated in 17 different wards but primarily from the pediatric intensive care unit (24.5% [24/98]), coronary care unit (CCU) (14.3% [14/98]), and department of respiration (14.3% [14/98]). Most patients had pneumonia (67.4% [59/86]) or sepsis (20.2% [18/86]).

**Genome sequencing of *Klebsiella pneumoniae* strains.** The 98 isolated *K. pneumoniae* strains were subjected to short-read Illumina sequencing; all genomes were *de novo* assembled starting from reads. The multilocus sequence type (MLST) was determined by the assembled genome. Read mapping against *K. pneumoniae* reference sequence (GenBank accession number NZ_CP045263.1) showed an average of 94.59% genome coverage among all 98 genomes. A reference based whole-genome alignment was

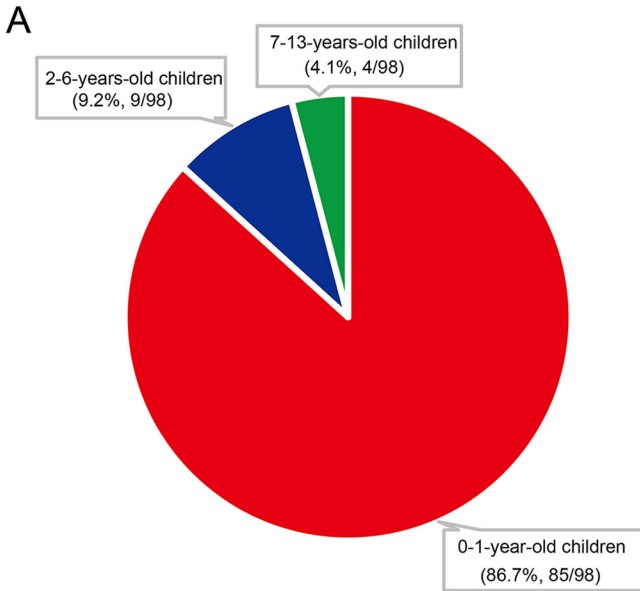

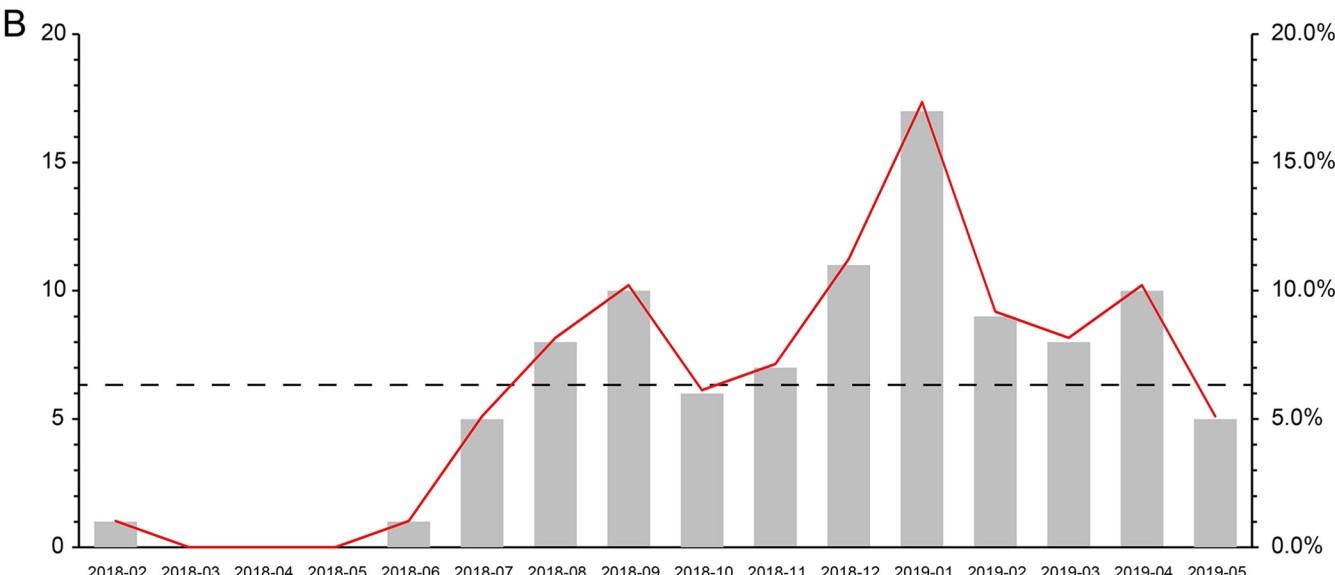

**FIG 1** Age composition and sampling date of each strain. (A) Pie chart representing the age composition of enrolled patients. (B) Histogram shows the number of *K. pneumoniae* strains isolated per month from February 2018 to May 2019. The dashed line corresponds to the mean ratio of *K. pneumoniae* strains. The plot (red line) shows the monthly ratio of *K. pneumoniae* strains (secondary *y* axis).

constructed, from which 143,674 single nucleotide polymorphisms (SNPs) were identified. The pairwise disparity between strains ranged from 0 to 39,815 SNPs. Putative recombination loci were further detected and removed, and 57,242 variable SNP sites were identified. This recombination-free SNP variation across the whole genome was used to construct a maximum likelihood phylogenetic tree of all 98 sequenced *K. pneumoniae* strains (Fig. 2A) and revealed the presence of a major clade of ST11.

The 98 *K. pneumoniae* strains belonged to 15 sequence types (Fig. S1). Seventy-two ST11 strains (73.5% [72/98]) were found in 13 different wards (76.5% [13/17]) from 8 different sample types (88.9% [8/9]), indicating that ST11 strains were dominant and widely distributed throughout the hospital. The remaining 26 strains belonged to ST76 (8.2% [8/98]), ST20 (3.1% [3/98]), ST193 (2.0% [2/98]), ST716 (2.0% [2/98]), ST1140 (2.0% [2/98]), and 9 other singular STs. A novel sequence type, ST5923 of strain xz084, was identified and was closest to the known ST252, with only one mutation in the *infB-5* allele.

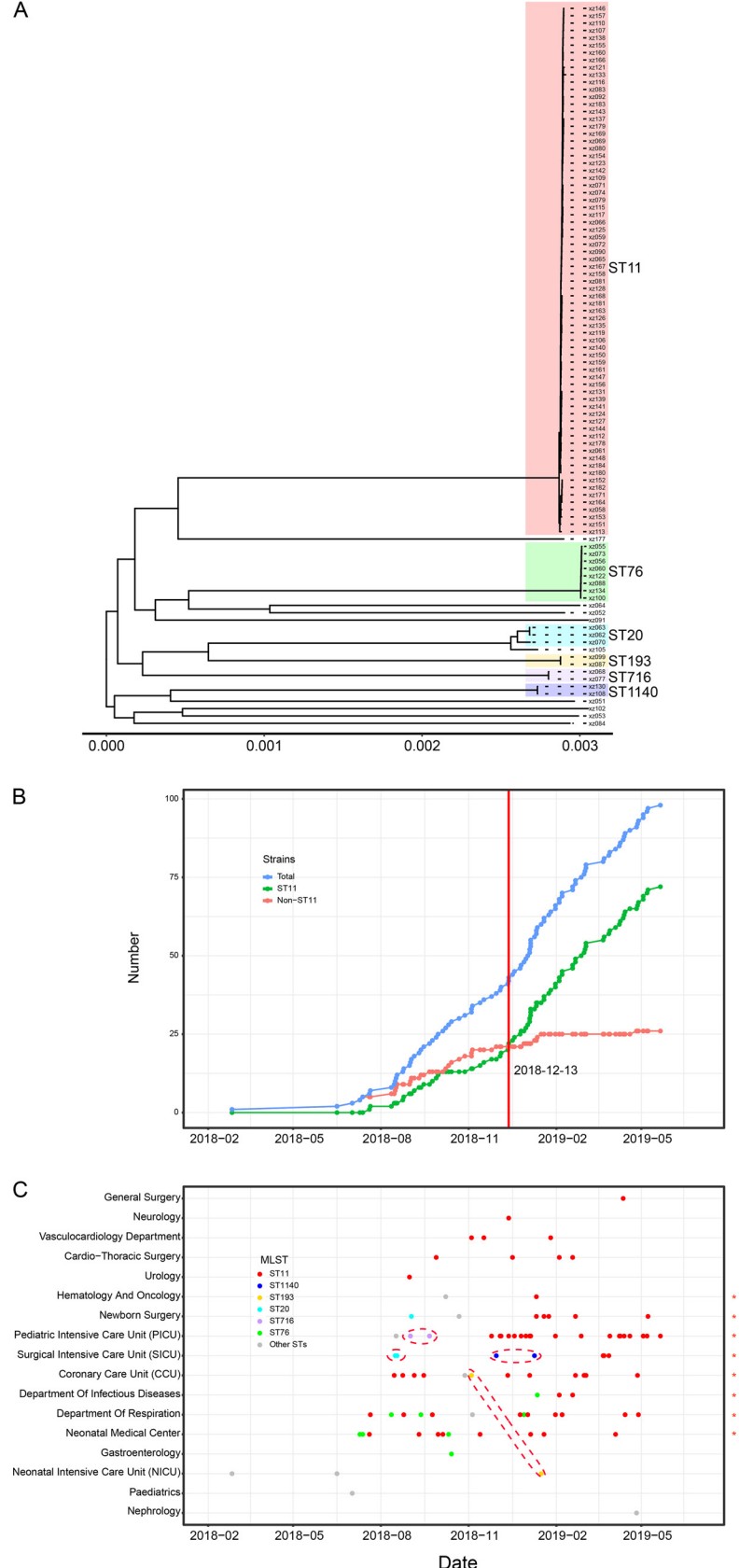

**FIG 2** Phylogeny and spatiotemporal distribution of strains. (A) Maximum likelihood phylogeny of 98
*K. pneumoniae* strains based on the 3,790,951-bp recombination-free alignment consisting of 57,242

**Emergence and rapid clonal spread of ST11 strains.** The temporal distribution pattern of collected strains revealed a clonal spread of ST11 strains. The collection of *K. pneumoniae* strains began on 26 February 2018, and most obtained before 13 December 2018 were not ST11 strains. On 20 July 2018, the first ST11 strain was isolated from a 1-month-old child; thereafter the number of ST11 strains increased rapidly, and ST11 strains eventually became the predominant strains in the hospital (Fig. 2B). The spatial distribution pattern revealed that the ST11 strains first emerged in the neonatal medical center and then gradually spread to other wards (Fig. 2C). Interestingly, among the eight wards with ST11 and other ST strains, the last collected stains were all ST11; the other STs were all isolated only during the early stage of collection. Both the temporal and spatial distributions suggested a nosocomial outbreak and clonal spread of ST11 strains.

Apart from the major clonally disseminated ST11 strains, there were also 5 minor STs with two or more strains (Fig. 2A). The pairwise number of SNPs within each ST was calculated and showed that all these minor ST strains were also highly clonal (Fig. 2C and Fig. S2). Eight SNPs were identified between the two ST716 strains which were both isolated from the pediatric intensive care unit (PICU) 20 days apart. In the surgical intensive care unit (SICU), 12 SNPs were identified between the two ST20 strains isolated 2 days apart, and 6 SNPs were identified between the two ST1140 strains isolated 40 days apart. While the two ST193 strains were isolated from 2 different wards and 73 days apart, only six SNPs were identified. Among the eight ST76 strains isolated in 4 different wards from July 2018 to January 2019, SNPs ranging from 2 to 15 were identified.

**Defining the outbreak clade.** To further analyze the highly spatiotemporally associated ST11 strains, a Bayesian phylogenetic tree based on the nonrecombination alignment was constructed. Phylogenetic analyses resolved the 72 ST11 strains into two major clades together with 6 strains falling outside the clades (Fig. 3A). Clade 1 comprised 62 strains and clade 2 consisted of 4 strains. The pairwise number of SNPs within clade 1 strains ranged from 0 to 29 SNPs (median, 6), which was comparable to the variation seen in clade 2 (2 to 13 SNPs; median, 7.5); 121 to 139 SNPs (median, 127) were found between clade 1 and 2 strains. Pairs of genomes within a distance less than 30 SNPs (29, 30) were considered the same transmission cluster (Fig. 3B and Fig. S3A). Two distinct transmission clusters were identified corresponding to 62 clade 1 and 4 clade 2 strains. A strong temporal signal was found in the accumulation of mutations on the nonrecombining fraction of the chromosomal sequences of the clade 1 strains (Fig. S3B), justifying the use of tip-dating phylogenetic calibration. The most recent common ancestor analysis indicated that these clade 1 strains shared a common ancestor in June 2017 (2017-06; 95% highest posterior density [HPD], 2016-05 to 2018-03) with an estimated evolutionary clock rate of $1.28 \times 10^{-6}$ ($8.08 \times 10^{-7}$ to $1.82 \times 10^{-6}$) substitutions per site per year (Fig. 3C). This inferred time predates the admission of the first patient (February 2018) by over 8 months.

**Tracking the transmission and evolution of outbreak strains.** The dated tree of the 62 clade 1 strains was then used to generate a transmission tree using the TransPhylo package. Five subclades identified in the dated tree were highly consistent with the diffusion pathways inferred from the transmission tree (Fig. 3D). The resulting structure showed a star-like topology centered around strain xz061 isolated in the CCU, suggesting a nonlinear spread of infection. Multiple transmission events probably took place, all starting from strain xz061 and spreading to other children in the hospital. Eighteen conceivable transmission links were identified within the 62 clade 1 strains based on transmission probabilities ($P \geq 0.5$). Excluding the 4 links between strains isolated from the same patient, 14 possible interpatient transmission links were identified (Fig. 3E). Three transmissions links were inferred from patients within the

**FIG 2 Legend (Continued)**
variable SNPs. (B) Cumulative numbers of the total, ST11, and other ST *K. pneumoniae* strains. (C) Distribution of ST11 and other STs among wards, with each point representing an isolated strain. Eight wards yielding ST11 and other STs are marked by asterisks. Dashed lines indicate strains, other than ST11 and ST76, with less than 30 pairwise SNPs.

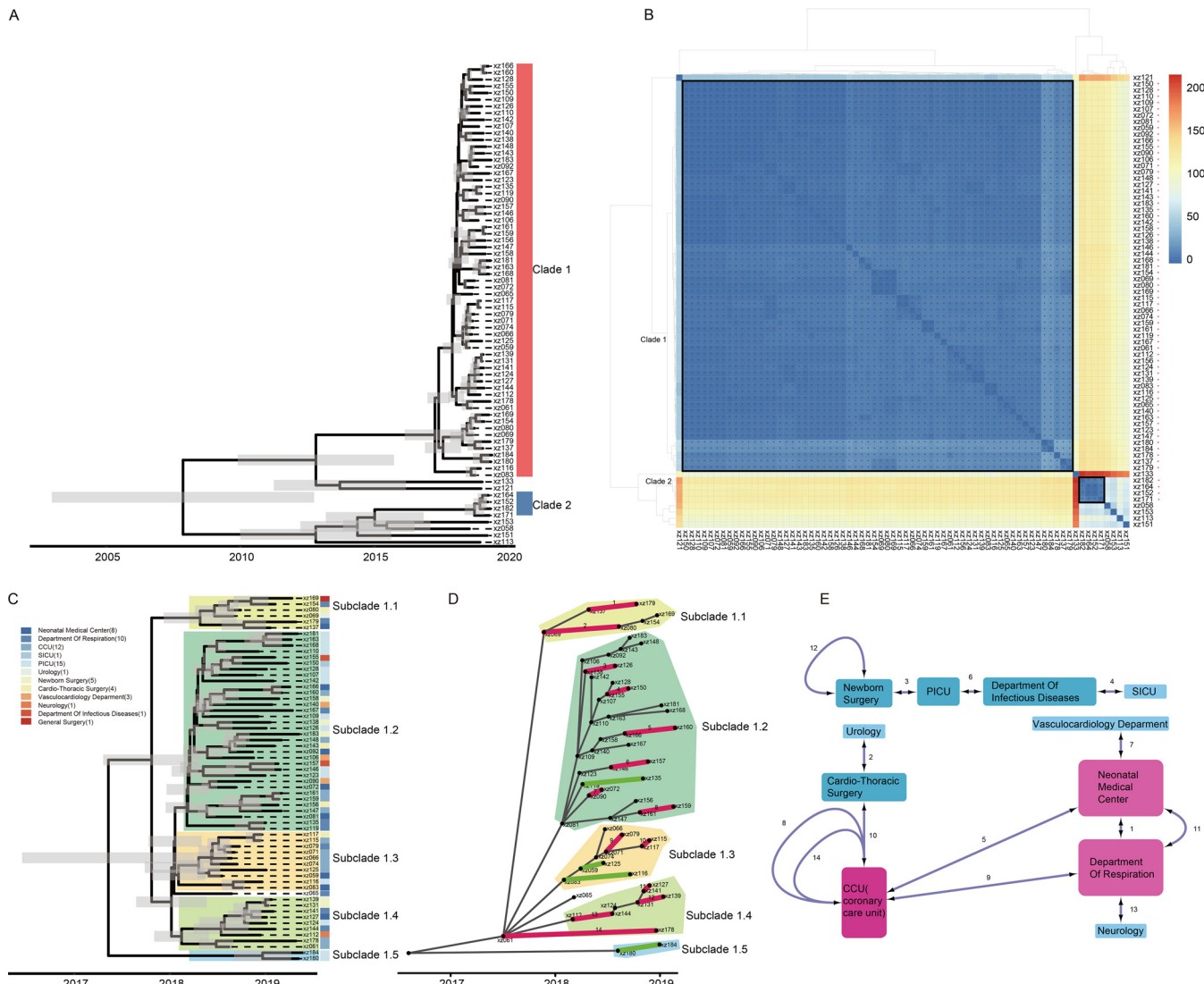

**FIG 3** Bayesian phylogeny and transmission of outbreak strains. (A) Bayesian phylogenetic tree based on the 5,383,824-bp nonrecombination alignment comprising 491 SNPs of 72 ST11 strains. Clades are defined as 1 and 2, with six outlying strains. (B) Pairwise SNP differences (number), as given by color scale on the top right. Pairs of genomes within a distance of 30 SNPs (SNP threshold) are marked by asterisks. (C) Dated phylogeny of the recombination of free chromosomal alignment of 62 clade 1 strains. Gray bars give the inferred 95% highest posterior density interval around node heights. (D) TransPhylo transmission tree. Probabilities of transmission links greater than 0.5 are thickened. Strains from different patients are colored red, while strains from the same patient are colored green. The x axis provides the mean time of infection. (E) Conceivable transmission links among patients from different wards. Link numbers were consistent with the TransPhylo transmission tree.

same ward, while 11 transmission links were identified among different wards. It is notable that 9 of the above 14 interpatient transmission links were associated with the CCU, the neonatal medical center, and the department of respiration. Only 5 links were identified among the remaining six wards.

To investigate the overall genome mutations of clade 1 strains, both the recombination and point mutations were analyzed; among the 152 SNPs identified within the 62 clade 1 strains, only 9 SNPs were located in the recombination region. The ratio of effects of recombination and mutation was 0.06 (9/143), indicating that mutation was the main evolutionary force of clade 1 strains during transmission. Tajima's D values were calculated per gene for clade 1 strains. Most of the genes showed negative Tajima's D values (Fig. S4), in particular 5 genes with more than one SNP, suggesting a possible negative selection or a recent population expansion. Four genes were identified with two nonsynonymous mutations, including the *acrE* gene, encoding the resistance nodulation cell division (RND) transporter periplasmic adaptor subunit belonging to the multidrug efflux

system; $ybtE$, encoding yersiniabactin biosynthesis salicyl-AMP ligase; $uvrY$, encoding the NarL family invasion response regulator; and $pqqF$, encoding the pyrroloquinoline quinone biosynthesis protein. Three synonymous mutations were identified in the gene encoding the type VI secretion system (T6SS) immunity phospholipase A1-binding lipoprotein Tli1-KP, which has been associated with antibacterial function (31).

**Antimicrobial resistance gene content and plasmid carriage of the outbreak isolates.** Aiming to explore the relationship between the clonal spread and antimicrobial resistance of the outbreak strains, antibiotic susceptibilities of all strains were determined, and the strains showed multiple-drug resistance to aminoglycosides, $\beta$-lactams, and fluoroquinolones (Table S2). Based on the assembled genomes, we predicted all the known resistance genes found within the sequences, and we observed a significant difference ($P < 0.001$) in the carbapenem resistance gene content between the ST11 and other ST strains (Fig. S5). The vast majority of the ST11 strains (95.8% [69/72]) carried the $bla_{KPC-2}$ gene. However, of the 26 non-ST11 strains, only 2 strains (7.7%) carried the $bla_{KPC-2}$ gene, while 14 strains (53.8%) carried $bla_{NDM-1}$, 5 strains (19.2%) carried $bla_{NDM-5}$, and 3 strains (11.5%) carried $bla_{IMP-4}$. The almost complete absence of these genes in the ST11 strains might be due to the absence of the plasmid carrying them. All ST11 strains carried the ColRNAI and IncFII plasmid replicons, which appear to be associated with the presence of several resistance genes, including $bla_{KPC-2}$, $rmtB$, $bla_{TEM-1B}$, $bla_{SHV.11\_1}$, $bla_{SHV.155\_1}$, $fosA\_3$, $oqxA\_1$, and $oqxB\_1$ (Fig. S5). However, only three genes ($bla_{KPC-2}$, $rmtB$, and $bla_{TEM-1B}$) among them were found on plasmid contigs. In addition, two clade 1 strains cocarried the $bla_{KPC-2}$ gene and $mcr-9$ gene, which were located on IncFII plasmid pHN7A8.1 and IncHI2/IncHI2A plasmid, respectively. IncHI2/IncHI2A and pKPC.CAV1321_1 plasmid replicons were found only in the three $mcr-9$-positive strains. Seven of the eight ST76 strains carried the IncX3_1 plasmid replicon, which appears to be associated with the presence of several resistance genes, including $bla_{NDM.1\_1}$, $bla_{SHV.12\_1}$, $fosA\_3$, $oqxA\_1$, and $oqxB\_1$ (Fig. S5). However, only the $bla_{NDM.1\_1}$ gene was found on plasmid contigs. The plasmid sequences were further explored using long-read sequencing assemblies. Five and four strains were selected from clade 1 and clade 2, respectively (Fig. 4A and Fig. S7). Plasmids of the remaining strains were identified by mapping reads to the complete plasmids assembled in the nine strains. All clade 1 and clade 2 strains carried two ColRNAI plasmids (11,970 and 5,596 bp, respectively); both had been reported in previously published ST11 genomes. Clade 1 strains carried an ~120,727-bp (IncFII) plasmid where the $bla_{KPC-2}$ gene was identified. While clade 2 strains also carried a $bla_{KPC-2}$ (IncFII/IncR) plasmid which shared large genomic regions with the $bla_{KPC-2}$ plasmid of clade 1 (Fig. S6), clade 2 strains also carried an ~219,804-bp (IncHI1B/repB) virulence plasmid and an ~87,088-bp (IncFII) resistance gene plasmid. Three clade 1 strains (xz163, xz168, and xz181) had acquired a plasmid (IncHI2/IncHI2A) with coexistence of the $mcr-9$, $mph$ (A), and $bla_{SFO-1}$ genes. Notably, all of these three strains were isolated from the pediatric intensive care unit and clustered closely in both evolutionary and transmission trees, which probably indicates that strain xz163 had evolved from a distant parent, xz061, containing 3 plasmids and then acquired a 4th plasmid which coharbored three resistance genes and continued to spread to the other two patients in the same ward (Fig. 3D and 4B).

## DISCUSSION

In this study, we focused on children under 1 year old and performed a high-resolution genomic analysis of nosocomial infection of multidrug-resistant *K. pneumoniae*, aiming to gain insight for controlling infection of children in the hospital. Nosocomial infection is an important cause of infant death (17), and multidrug-resistant *K. pneumoniae* has become an important pathogen causing nosocomial infection in children. The proportion of infants infected with multidrug-resistant bacteria was higher in low- and middle-income countries. Previous studies mainly focused on adults in developed countries, especially the elderly. To achieve the Millennium Development Goal for child

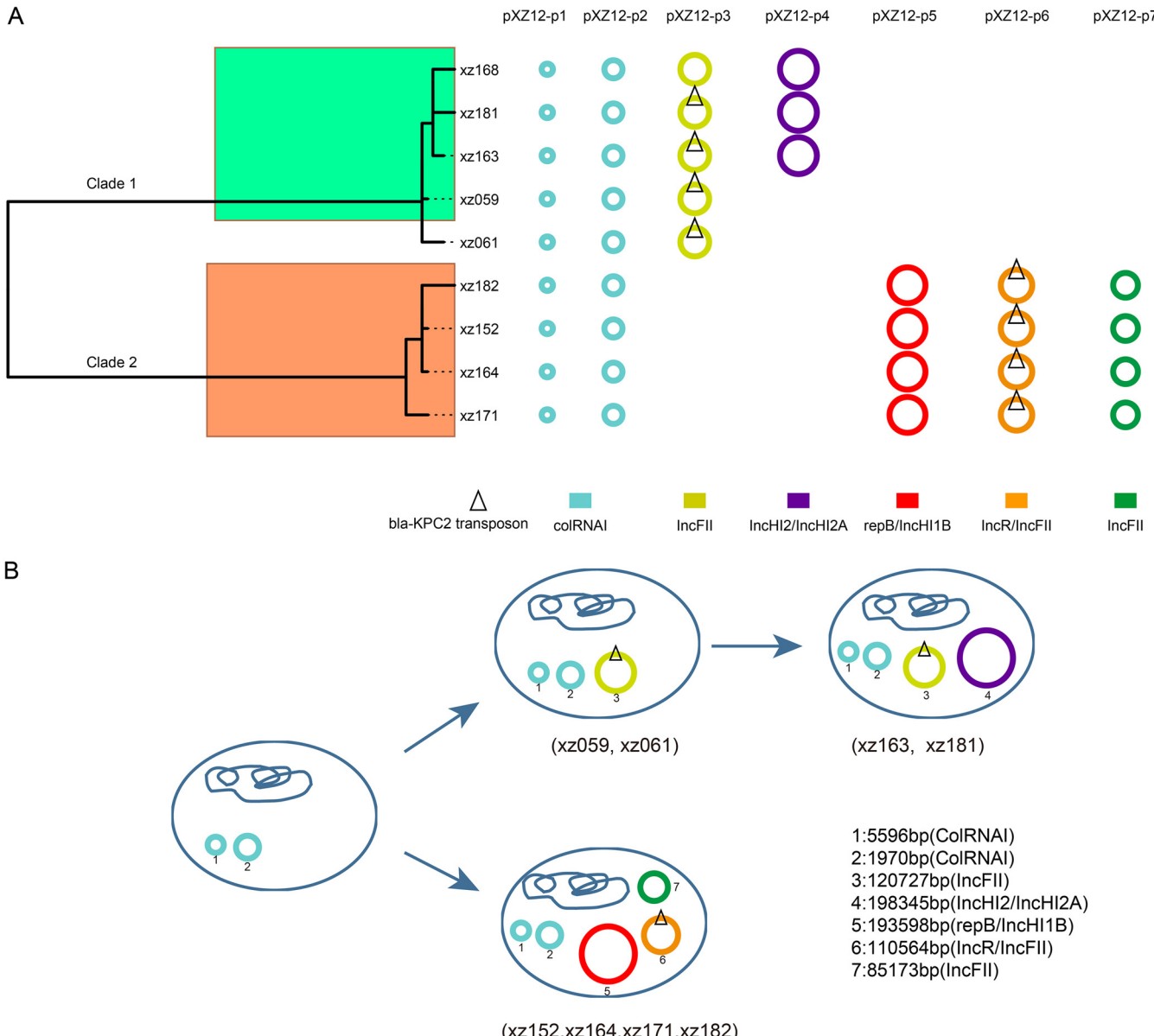

**FIG 4** Diversity in plasmid types and carriage across nine outbreak strains. (A) Plasmids carried by nine strains assembled using nanopore long-read technology. The color provides the Inc type assignment of the plasmids based on the presence of plasmid replicon sequences and is defined in the key. *bla*~KPC-2~-carrying plasmids are indicated by the triangular symbol. Virulence plasmids are defined as those carrying the *rmpA*, *rmpA2*, and/or *iuc* operon genes. The phylogeny on the left provides the core genome phylogeny for these nine strains, with chromosomal clades 1 and 2 highlighted in color. (B) Schematic representation of plasmid acquisitions and evolution of ST11 strains. Two common plasmids were shared by 2 clades. An IncFII KPC-2 positive plasmid was acquired by clade 1 strains, followed by a large *mcr-9*-positive plasmid. One virulence plasmid and two resistance plasmids were acquired by clade 2 strains.

survival of the United Nations, the key point is to reduce the infection rate of children in low- and middle-income countries.

By sequencing the whole genomes of 98 strains, ST11 was identified as the dominant MLST in the hospital, which is consistent with previous results (32). Apart from the ST11 strains, many other types of *K. pneumoniae* strains coexist in the hospital; 14 different MLSTs were found, of which ST5923 was a novel type. In addition to ST11, there were 5 MLSTs isolated with two or more strains. The strains within the same types were highly clonal, and the isolation time and sampling wards were highly overlapped. These data showed that besides ST11, many other MLSTs of *K. pneumoniae* can also cause infections and spread to a certain extent in the hospital under appropriate conditions.

Based on the spatiotemporal analysis, it was found that ST11 was not the most abundant ST at the beginning of the collection period, but it increased gradually, becoming the most common ST in collected strains. Several other ST strains had also spread to a small extent in the early stage, but in the end, the ST11 strains competed to arise and spread widely to many other wards, showing stronger transmission ability.

Within the 72 ST11 strains, two different clades were identified. Apart from the two common ColRNAI plasmids, clade 1 strains acquired one IncFII type KPC-2 plasmid, while clade 2 strains obtained three other plasmids, two drug-resistant plasmids and one virulence plasmid. Transmission analysis results showed that clade 1 strains had infected more children and spread to more wards than clade 2 strains (results not shown). We speculated that the acquired plasmids had different effects on the transmissibility and pathogenicity of the host strains (33). Clade 1 strains might have lower plasmid burden than clade 2 strains, thus exhibiting a competitive advantage and stronger transmissibility. Although clade 2 contained only 4 strains, they might pose a greater threat to hospitalized children by coharboring both resistance plasmids and a virulence plasmid. Control measures should be implemented to avoid further spread of such strains in the hospital environment.

During the clonal spread of *K. pneumoniae* strains, they are still evolving to adapt to the complex hospital environment. Chromosomal mutations associated with virulence and resistance genes were identified within clade 1 strains. Three clade 1 strains isolated from the pediatric intensive care unit had acquired a gene *mcr-9*-positive plasmid during transmission. ST11 *K. pneumoniae* coharboring the $bla_{KPC-2}$ and *mcr-1* gene plasmid was first reported (34) in the same province as this study in 2016, nearly 2 years before the current collection, indicating the persistent existence and spread of the colistin resistance plasmids in hospital strains.

The application of whole-genome sequencing in this study facilitated deeper insight into the spread of strains between patients and identified possible transmission events. Genomic analysis revealed that *K. pneumoniae* circulated for at least 8 months before the initiation of our sampling efforts. The coronary care unit, neonatal medical center, and the department of respiration were the foremost environments for direct patient-to-patient transmission. Autonomous movement during hospitalization is unlikely for children under 1 year of age, suggesting possible roles of staff and/or shared equipment in transmission. Identification of particular transmission events may inform targeted interventions to prevent CRKP nosocomial infections in pediatric patients.

## MATERIALS AND METHODS

**Bacterial isolation and antimicrobial susceptibility testing.** From February 2018 to May 2019, a total of 98 *K. pneumoniae* strains were isolated from pediatric patients in different wards in the Affiliated Hospital of Xuzhou Medical University, China. The bacterial strains were isolated from clinical specimens, which included sputum, urine, blood, and other samples. Strain identification was performed by matrix-assisted laser desorption ionization–time of flight mass spectrometry. *In vitro* antimicrobial susceptibility testing of isolates was analyzed with a Vitek-2 compact system (bioMérieux, Marcy-l'Etoile, France). Antimicrobial susceptibility testing was interpreted in accordance with the Clinical and Laboratory Standards Institute (CLSI), except for tigecycline and colistin, for which results were interpreted based on the European Committee for Antimicrobial Susceptibility Testing (EUCAST) criteria.

**Whole-genome sequencing.** The DNA library was prepared at the sequencing company. In brief, genomic DNA was extracted from bacterial colonies using a QIAamp DNA minikit (Qiagen, Hilden, Germany) and quantified using a Qubit fluorometer (Life Technologies, Carlsbad, CA, USA). The libraries were constructed using the Nextera XT kit (Illumina Ltd., San Diego, CA, USA) according to the manufacturer's recommendations and sequenced in pair-end mode (2 × 150 bp) using the Illumina HiSeq 2500 platform at Novogene Company (Beijing, China) with a median sequencing depth of 235.6×. The nanopore sequencing library was prepared using the SQK-RAD004 rapid sequencing kit (Oxford Nanotechnology, UK) and sequenced on a MinION Mk1B R9.4 flow cell in our lab.

**Bioinformatics analyses.** Sequencing reads were quality filtered using FastQC v0.11.8 software; adaptors and low-quality reads were removed and filtered out using Trimmomatic with default parameters. We ran the genomic distance estimation tool Mash v2.3 (35) on the trimmed and quality-filtered Illumina raw sequencing reads to identify the best-matching chromosomal reference against an archive of complete *K.*

*pneumoniae* genomes from NCBI RefSeq. The best-matching chromosomal reference was *K. pneumoniae* strain 16HN-263 (GenBank accession number NZ_CP045263.1). Reads were mapped against this reference using bwa-mem v0.7.17 (36). Hybrid assembly of Illumina and nanopore sequencing reads was carried out with Unicycler v0.4.8 (37) in normal mode. Multilocus sequence typing, resistance, virulence, and plasmid profiles were characterized starting from the final assemblies using resfinder, vfdb, and plasmidfinder databases in ABRicate v0.9.7 and Kleborate v2.2.0. (38) We also analyzed our short sequencing reads using SRST2 software v0.2.0 (39). Novel alleles of ST profiles were submitted to BIGSdb v1.32.0 (40).

Initially, SNPs were predicted using Snippy v4.6.0; the consensus whole-genome alignment was used to infer a maximum likelihood phylogenetic tree using Raxml-ng v1.0.3 (41), implementing the GTR+ model with 1,000 bootstrap replicates. The whole-genome alignment and maximum likelihood phylogenetic tree were combined to identify the recombination regions using ClonalFrameML v1.12 (42). Then we excluded the identified recombination regions and left a 5,383,824-bp nonrecombination alignment comprising 491 SNPs across the 72 ST11 strains. A maximum likelihood phylogenetic tree based on the nonrecombination alignment was constructed using Raxml-ng v1.0.3 implemented with 1,000 bootstrap replicates. GTR+ was selected as the best evolutionary model by using Modeltest-ng v0.1.7.

The distribution of the pairwise SNP distances among strains was visualized as shown in Fig. S3A in the supplemental material; the SNP cutoff threshold was determined manually by looking at the graph and finding that there were two distinctive groups of genome pairs, those with less than 30 SNPs and those with more than 43 SNPs, which allowed us to safely infer that two genomes could be considered part of the same transmission cluster if their distance in number of SNPs was lower than the 30-SNP threshold value.

We used a root-to-tip regression of sampling dates against genetic diversity in TempEst v1.5.3 (43), optimizing the best fit for the root to maximize the determination coefficient $R^2$.

Following the identification of a strong temporal signal in the clade 1 strains, we used a Bayesian evolutionary analysis to infer a timed phylogeny by sampling trees with BEAST v2.4.7 by using the concatenated nonrecombination alignment and labeled the time tips of each strain using the sampled date. We used the GTR substitution model with a coalescent constant population size and a strict molecular clock rate. The model was run using a Markov chain Monte Carlo (MCMC) chain length of 20,000,000 with a 10% burn-in; convergence of the chain was inspected using Tracer v1.7.2. A maximum clade credibility tree was generated using TreeAnnotator v2.4.7 and visualized using Figtree v1.4.2.

This timed phylogeny tree was then used as input for the transmission tree inference using the R package TransPhylo (44) v1.4.5. A gamma distribution was calculated with a shape parameter of 1.2 and a scaling parameter of 1.0 (45) specified as priors for the generation time. The MCMC was run for 10 million iterations to ensure convergence. The genome annotation was performed using the NCBI Prokaryotic Genome Annotation Pipeline. Plasmid images were produced in CGView Comparison Tool v1.0.3 (46) and Easyfig v2.2.2 (47). Pearson's chi-squared test was performed for the strains carrying the $bla_{KPC-2}$ gene between ST11 and non-ST11 multilocus sequence types; the $\chi^2$ squared and *P* value were calculated to infer statistically significant difference.

**Ethics approval and consent to participate.** A guardian of each child patient provided written informed consent for study participation before enrollment. This study was conducted in accordance with the Declaration of Helsinki. The Clinical Research Ethics Committee of the Affiliated Hospital of Xuzhou Medical University approved the study (XYFY2015-JS016-01), as all samples evaluated in this study were initially collected for diagnosis during patient care and were thereby obtained without increasing the patients' medical costs and suffering.

**Data availability.** All data generated or analyzed during this study are included in this published article and its supplemental material.

Whole-genome sequences of this project have been deposited in GenBank under accession numbers JAHQME000000000 to JAHQQZ000000000 and BioProject number PRJNA739673.

## SUPPLEMENTAL MATERIAL

Supplemental material is available online only.

**SUPPLEMENTAL FILE 1**, PDF file, 5.5 MB.

## ACKNOWLEDGMENTS

This study was supported by grants from the National Major Science and Technology Project (no. 2021YFC2301002), National Natural Science Foundation of China (31900151), and a Research Foundation for Advanced Talents of Guangdong Provincial People's Hospital (no. KJ012021097).

K.W., J.L., Q.C., and Y.L. performed experiments, P.L., H.S., and B.G. provided methods or reagents for this work. J.L., B.T., and B.G. provided clinical input and data for this work. X.L., J.C., and P.L. performed analysis for this work. X.L., J.C., and P.L. wrote the first draft of the manuscript. X.L., K.W., J.C., J.L., J.L., Q.C., Y.L., B.T., H.S., P.L., and B.G. contributed to and read the final version of the manuscript.

We declare that we have no competing interests.

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
