## [Reviewer comments · Microbiology Spectrum]

Microbiology Spectrum

Clonal spread of carbapenem-resistant *Klebsiella pneumoniae* ST11 in Chinese pediatric patients

Xiong Liu, Kaiying Wang, Jiali Chen, Jing-Wen Lyu, Jinhui Li, Qichao Chen, Yanfeng Lin, Benshun Tian, Hongbin Song, Peng Li, and Bing Gu

Corresponding Author(s): Peng Li, Chinese PLA Center for Disease Control and Prevention

Review Timeline:

Submission Date:	May 24, 2022
Editorial Decision:	June 24, 2022
Revision Received:	September 26, 2022
Editorial Decision:	October 19, 2022
Revision Received:	October 24, 2022
Accepted:	November 4, 2022

Editor: Maria De Francesco

Reviewer(s): Disclosure of reviewer identity is with reference to reviewer comments included in decision letter(s). The following individuals involved in review of your submission have agreed to reveal their identity: Erika Scaltriti (Reviewer #2); Alaa Abouelfetouh (Reviewer #3)

Transaction Report:

DOI: <https://doi.org/10.1128/spectrum.01919-22>

June 24, 2022

Dr. Peng Li
Chinese PLA Center for Disease Control and Prevention
DONGDA street 20#
Beijing
China

Re: Spectrum01919-22 (Clonal spread of carbapenem-resistant *Klebsiella pneumoniae* ST11 in Chinese pediatric patients)

Dear Dr. Peng Li:

Link Not Available

Sincerely,

Maria De Francesco

Journals Department
Reviewer comments:

Reviewer #1 (Comments for the Author):

Extensive work done.

Kindly review the methodology on "Bacterial isolation and antimicrobial susceptibility testing".

What were the hand hygiene audit results for the wards?

Reviewer #2 (Comments for the Author):

Liu and colleagues describe the clonal spread of carbapenem-resistant *Klebsiella pneumoniae* ST11 in Chinese pediatric patients through an extended genomic analysis including phylogeny and mobile element content (antibiotic resistance and virulence genes). Results, in particular the use of phylodynamics in outbreak context, are interesting and this paper is an example of the use of multiple genomic tools to understand outbreak dynamics, in particular for *Klebsiella pneumoniae* in children hospital context. Apart from these aspects, many formulations and statements are inadequate (see section "Technical and content comments").

Regarding the style, the manuscript would benefit from review by a native speaker.

Reviewer #3 (Comments for the Author):

The study investigates the population structure and phylogeny of 98 *Klebsiella pneumoniae* isolates obtained from pediatric patients in a Chinese hospital between February 2018 and May 2019. The authors reported the clonal spread of KPC-2 producing ST11 *K. pneumoniae* strains that were clustered into 2 clades. The study described the plasmid content in the 2 clades complete with resistance genes. The authors used these findings to explain the higher transmissibility among clade 1 isolates relative to clade 2 isolates. The study calls for a wider use of genomic tools to identify outbreaks.

Minor comments were observed:

The manuscript needs minor English revision and editing.

-Line 64: Would the authors please explain the importance of carbapenem as last resort agent in the treatment and hence the global concern as a result of carbapenem resistance development?

-Line 117: was the DNA library prepared at the sequencing company? Please give some details about Illumina library prep.

-Line 124-125: Would the authors please give the details of quality check and trimming of the reads?

-Would the authors give more details about Fig. 3c as it is not clear how they inferred the substitutions per site per year from the figure.

-Lines 297-297: Fosfomycin and macrolide susceptibility data are not shown in the table.

-Line 300: Do the authors mean a statistically significant difference?

-Lines 304-306: The authors would better comment on the almost complete absence of these genes in the ST11 strains.

-Line 306: The authors would better specify that they mean IncFII.pHN7A8_1

-Line 306-308: "All ST11 strains carried the ColRNAI and IncFII plasmid replicons, which were highly correlated with the presence of blaKPC-2, rmtB, and blaTEM-1B genes" the isolates also carried blaSHV.11_1, blaSHV.155_1 as well as fosA_3 coding for Fosfomycin resistance and oqxA_1 and oqxB_1 coding for fluoroquinolone resistance.

-Line 308: Looks like only 2 clade 1 strains (and not 3) co-carried blaKPC-2 and mcr-9.

-Line 310: Looks like the three mcr-9 positive strains carried the pKPC.CAV1321_1 and not repB_KLEB_VIR plasmid replicon.

Lines 311-313: the same can be said about ST76 strains and IncX3_1 plasmid replicon and blaNDM.1_1, blaSHV.12_1, fosA_3, oqxA_1 and oqxB_1.

Would the authors please comment on Fig 5B at the end of the results? From Fig 3D and Fig 5B, it seems that strain xz163 evolved from a distant parent xz061 and acquired a 4th plasmid then further evolved to xz168 and xz181.

Lines 314-316: Please mention Additional file 8 Figure S6 here.

Lines 371-372: Figures 3D and 3E show the transmission of clade 1 but not clade 2, so unless the authors are referring to other figures here, would they just mention that these results are not shown?

Staff Comments:

Preparing Revision Guidelines

Please return the manuscript within 60 days; if you cannot complete the modification within this time period, please contact me. If you do not wish to modify the manuscript and prefer to submit it to another journal, please notify me of your decision immediately so that the manuscript may be formally withdrawn from consideration by Microbiology Spectrum.

Technical and content comments:

Line 71-72: blaKPC-2 is not a genotype. blaKPC-2, among antibiotic resistance genes, is largely diffused or the most diffused

Line 110-115: the phrase should be simplified

Line 118: the median value should be calculated and indicated

Line 119-121: type of flow cell and MinION should be added

Line 124: how reads are quality filtered? Please specify Q30 cut off and read length cut off (eventually)

Line 130 and subsequent: Version of software should be added after software name, first of references.

Line 130bis: for hybrid assembly, add details on the usage of Unicycler (e.g. conservative mode)

Line 132: Specify databases used with Abricate and if you are starting from reads or assembly (it changes sometimes results)

Line 134 and 138: What SNP pipeline do you use as input for phylogenesis? Is not clear if snippy or Roary. Please clarify.

Line 143-147: Add some information on used models (type of molecular clock and population). Add software name (BEAST?)

Line 149: "Whole genome shotgun project" is not appropriate. Use "whole genome sequences of this project"

Line 175: substitute "performed with" "subjected to"

Line 176: "all reads were de novo assembled" substitute "all genomes were de novo assembled starting from reads"

Line 177: substitute "Read mapping to" with "Read mapping against"

Line 183: the final size of the alignment (3790951 bp) is not necessary (see also below)

Line 188: "a structure dominated by" is not entirely correct. Refer to the presence of a major clade of ST 11

Line 226: substitute with "SNPs ranging from 2 to 15 were identified"

Line 229-230: same as Line 183

Line 238: how do you choose this threshold? Do you have references? If yes, please add it. It's a very high number of SNPs and this high threshold may be misleading. This is a very IMPORTANT aspect

Line 241: how do you calculate that temporal signal was strong? (e.g. Tempest software)

Line 270: is not clear if you are working on a non-recombinant dataset or not. Here you mentioned SNPs in recombinant region, but you removed them (line 182).

Line 277: clarify the reason why you mentioned these genes (these genes are under pressure in your opinion?). These observations are not well explained.

Line 289: replace "clades" with "isolates"

Line 304: "highly correlated" is not clear, explain better. blaKPC-2 is only in this type of plasmids. If yes, supply references

Line 311: replace “plasmid genomes” with “plasmid sequence”

Line 328: Add “probably” first of “indicate that...”

Line 347: remove “the” at the beginning of the line

Line 356-359: this phrase is not clear. Rewrite.

Line 364: the term “obtained” is not appropriate

Line 381-383: this concept is trivial. Rewrite or remove it

Line 389: replace with “plasmids in hospital strains”

Figure 2C: indicate the meaning of dotted lines in the caption

Figure 4: I suggest to visualise only genes that are present at least in one genome to simplify the figure. Maybe a completed figure can be put on supplementary.

**Response to the reviewers' comments:**

**Reviewer: 1**

**Point 1:**

Extensive work done.

Kindly review the methodology on "Bacterial isolation and
antimicrobial susceptibility testing".

What were the hand hygiene audit results for the wards?

**Response 1:**

Thanks very much for the reviewer's comments, we have revised the
methodology on "Bacterial isolation and antimicrobial susceptibility
testing" in the resubmitted manuscript. (Lines 104-113)

Hospital infection department had taken strict hand hygiene audit
measures to control the potential outbreak in the hospital. First,
pre-screening of carbapenem-resistant Enterobacteriaceae (CRE) in
sputum samples and rectal swabs were introduced before admission
to the neonatal medical ward. Second, strict isolation procedures
were implemented for patients with CRE infection. Third, it was
necessary for medical staff who contact with patients infected with
CRE to go through a disinfection procedure. Finally, the neonatal

medical wards where newborns with CRE infection stayed were
thoroughly sterilized after the discharge of the patients. The
sterilized ward left unoccupied for more than two weeks before new
patients were admitted.

**Reviewer 2:**

**Point 2:**

Liu and colleagues describe the clonal spread of
carbapenem-resistant *Klebsiella pneumoniae* ST11 in Chinese
pediatric patients through an extended genomic analysis including
phylogeny and mobile element content (antibiotic resistance and
virulence genes). Results, in particular the use of phylodynamic in
outbreak context, are interesting and this paper is an example of the
use of multiple genomic tools to understand outbreak dynamics, in
particular for *Klebsiella pneumoniae* in children hospital context.
Apart from these aspects, many formulations and statements are
inadequate (see section "Technical and content comments").
Regarding the style, the manuscript would benefit from review by a
native speaker.

**Response 2:**

Thanks very much for the reviewer's all constructive comments, we
have carefully revised our manuscript according to the reviewer's

comments and improved the manuscript by a native speaker. The
comments raised by reviewer 2 have been responded point by point
below.

**Technical and content comments:**

**Point 3**

**Line71-72: blaKPC-2 is not a genotype. blaKPC-2, among**
**antibiotic resistance genes, is largely diffused or the most**
**diffused.**

**Response 3:**

Sorry, it was a mistake, we have modified the description. “ST11 is
the most prevalent CRKP sequence type (ST) in China, and *bla*_{KPC-2}
is one of the most common drug resistance genes.” (Lines 70-72)

**Point 4:**

**Line 110-115: the phrase should be simplified.**

**Response 4:**

Thanks very much for the reviewer's comments, we have simplified
the phrase. “MICs were interpreted according to the CLSI (2021)
guidelines. The electronic medical records of culture-positive
children were reviewed retrospectively to obtain demographic and
clinical data.” (Lines 110-113)

**Point 5:**

**Line 118: the median value should be calculated and indicated.**

**Response 5:**

Thanks very much for the reviewer's comments, we have revised the
description in the resubmitted manuscript. "With a median
sequencing depth of 235.6X." (Lines 122-123)

We have calculated both the mean and median value of sequencing
depth, and the two values were very close, the mean sequencing
depth was 235.8X and the median sequencing depth was 235.6X.

**Point 6:**

**Line 119-121: type of flow cell and MinION should be added.**

**Response 6:**

Thanks very much for the reviewer's comments, we have added the
type of flow cell and MinION in the revised manuscript. "The
nanopore sequencing library was prepared using the SQK- RAD004
rapid sequencing kit (Oxford Nanotechnology, UK) and sequenced
on MinION Mk1B R9.4 flowcell in our lab." (Lines 123-126)

**Point 7:**

**Line 124: how reads are quality filtered? Please specify Q30 cut**

**off and read length cut off (eventually)**

**Response 7:**

Thanks very much for the reviewer's comments, we have described
how reads were quality filtered in the revised manuscript.

“Sequencing reads were quality filtered using the FastQC v0.11.8
software, adapters and low-quality reads were removed and filtered
out using Trimmomatic with default parameters.” (Line 128-130)

We also have specified the Q30 cut off and read length cut off in the
revised manuscript. “The Q30 cut off was set to 85% and the read
length cut off was set to longer than 100bp.” (Lines 130-131)

**Point 8:**

**Line 130 and subsequent: Version of software should be added**
**after software name, first of references.**

**Response 8:**

Thanks very much for the reviewer's comments, we have added the
version of software after the software name as suggested.

**Point 9:**

**Line 130bis: for hybrid assembly, add details on the usage of**
**Unicycler (e.g. conservative mode)**

**Response 9:**

Thanks very much for the reviewer's comments, we have added
details on the usage of Unicycler in the revised manuscript. “hybrid
assembly was carried out with Unicycler v0.4.8 in normal mode.”
(Lines 138-140)

**Point 10:**

**Line 132: Specify databases used with Abricate and if you are**
**starting from reads or assembly (it changes sometimes results)**

**Response 10:**

Thanks very much for the reviewer's comments, we have specified
the starting and the databases used with Abricate in the revised
manuscript. “Resistance, virulence and plasmid profiles were
characterized starting from the final assemblies using resfinder, vfdb
and plasmidfinder database in ABRicate” (Lines 140-143).

We also analyzed our short sequencing reads using the SRST2
software v0.2.0, which did not change and influence our final
results.

**Point 11:**

**Line 134 and 138: What SNP pipeline do you use as input for**
**phylogenesis? Is not clear if snippy o roary. Please clarify.**

**Response 11:**

Thanks very much for the reviewer's comments, we have added the
SNP pipeline used as input for phylogenesis in the revised
manuscript.

“Initially, SNPs across the whole genome were predicted using
Snippy v4.6.0, the consensus whole genome alignment was used to
infer a maximum likelihood phylogenetic tree using Raxml-ng
v1.0.3 implementing with 1000 bootstrap replicates, GTR+ was
selected as the best evolutionary model by using Modeltest-ng
v0.1.7” (Lines 147-151)

**Point 12:**

**Line 143-147: Add some information on used models (type of**
**molecular clock and population). Add software name (BEAST?)**

**Response 12:**

Thanks very much for the reviewer's comments, we have added the
software name and information on used models in the revised
manuscript.

“We used a Bayesian evolutionary analysis to infer a timed
phylogeny by sampling trees with BEAST v2.4.7 by using the
concatenated non-recombination alignment and labeled the time tips
of each strain using the sampled date. We used the GTR substitution

model with a coalescent constant population size and a strict
molecular clock rate.” (Lines 175-180)

**Point 13:**

**Line 149: “Whole genome shotgun project” is not appropriate.**

**Use “whole genome sequences of this project”**

**Response 13:**

Thanks very much for the reviewer's comments, the manuscript had
been revised as suggested.

“Whole genome sequences of this project have been deposited at
GenBank under accession numbers from JAHQME000000000 to
JAHQQZ000000000, BioProject ID: PRJNA739673.” (Lines
195-197)

**Point 14:**

**Line 175: substitute “performed with “subjected to”**

**Response 14:**

Thanks very much for the reviewer's comments, the manuscript had
been revised as suggested.

“The 98 isolated *K. pneumoniae* strains were subjected to short-read
Illumina sequencing” (Lines 221-222)

**Point 15:**

**Line 176: “all reads were de novo assembled” substitute “all**
**genomes were de novo assembled starting from reads”**

**Response 15:**

Thanks very much for the reviewer's comments, the manuscript had
been revised as suggested.

“All genomes were de novo assembled starting from reads.” (Lines
222-223)

**Point 16:**

**Line 177: substitute “Read mapping to” with “Read mapping**
**against”**

**Response 16:**

Thanks very much for the reviewer's comments, the manuscript had
been revised as suggested.

“Read mapping against *K. pneumoniae* reference sequence” (Line
224)

**Point 17:**

**Line 183: the final size of the alignment (3790951 bp) is not**
**necessary (see also below)**

**Response 17:**

Thanks very much for the reviewer's comments, the manuscript had

been revised as suggested.

“Putative recombination loci were further detected and removed, and

57,242 variable SNP sites were identified.” (Lines 229-230)

**Point 18:**

**Line 188: “a structure dominated by” is not entirely correct.**

**Refer to the presence of a major clade of ST 11**

**Response 18:**

Thanks very much for the reviewer's comments, the manuscript had

been revised as suggested.

“Revealed the presence of a major clade of ST11.” (Lines 233-234)

**Point 19:**

**Line 226: substitute with “SNPs ranging from 2 to 15 were**

**identified”**

**Response 19:**

Thanks very much for the reviewer's comments, the manuscript had

been revised as suggested.

“Among the eight ST76 strains isolated in 4 different wards from

July 2018 to January 2019, SNPs ranging from 2 to 15 were

identified.” (Lines 271-272)

**Point 20:**

**Line 229-230: same as Line 183**

**Response 20:**

Thanks very much for the reviewer's comments, the manuscript had
been revised as suggested.

“To further analyze the highly spatial-temporal associated ST11
strains, a Bayesian phylogenetic tree based on the
non-recombination alignment was constructed.” (Lines 274-276)

**Point 21:**

**Line 238: how do you choose this threshold? Do you have**
**references? If yes, please add it. It’s a very high number of SNPs**
**and this high threshold may be misleading. This is a very**
**IMPORTANT aspect**

**Response 21:**

Thanks very much for the reviewer's comments.

First, we have added description on how the threshold was chosen in
the revised manuscript.

“The distribution of the pair-wise SNPs distances among strains was
visualized as showed in the Additional file 5 Figure S3A, the SNP
cut-off threshold was determined manually by looking at the graph
and finding that there were two distinctive groups of genome pairs,
those with less than 35 SNPs and those with more than 35 SNPs,

which allowed us to safely infer that two genomes could be
considered as part of the same transmission cluster if their distance
in number of SNPs was lower than the 35 SNPs threshold value.”
(Lines 161-168)

Second, we also have found a reference in which pairs of genomes
within a distance of 35 SNPs were considered as part of the same
transmission cluster, and we have added it in the revised manuscript.

“Pairs of genomes within a distance of ≤ 35 SNPs (42) were
considered as the same transmission cluster.” (Lines 283-284)

**Point 22:**

**Line 241: how do you calculate that temporal signal was strong?**
**(e.g. Tempest software)**

**Response 22:**

Thanks very much for the reviewer's comments, we have added the
description on how the temporal signal was calculated in the revised
manuscript.

“We used a root to tip regression of sampling dates against genetic
diversity in TempEst v1.5.3, optimizing the best fit for the root to
maximize the determination coefficient R^2 . The slope of the
regression was positive, and the p-value is 0.0006, showing that the

genomic data reflect strong temporal signal.” (Lines 169-173)

**Point 23:**

**Line 270: is not clear if you are working on a non-recombinant**
**dataset or not. Here you mentioned SNPs in recombinant region,**
**but you removed them (line 182).**

**Response 23:**

Sorry for our unclearly descriptions, in the previous transmission
tree analysis part, we were working on the non-recombinant dataset,
the recombinant regions were removed. But in this part, all genome
mutations were analyzed including the recombination and point
mutation, thus, 9 mutations introduced by recombination and 143
mutations introduced by point mutation were described.

We have added “To investigate the overall genome mutations of
Clade1 strains, both the recombination and point mutations were in
analyzed (Lines 315-316) in the revised manuscript.

**Point 24:**

**Line 277: clarify the reason why you mentioned these genes (this**
**gene are under pressure in your opinion?). These observations**
**are not well explained.**

**Response 24:**

Thanks very much for the reviewer's comments.

We have changed “In addition, we identified 5 genes with more than
one SNP.” with “In this case, a recent population expansion was
more likely to happen. All the mutations were annotated according to
their genome positions, from which 5 genes were identified with
more than one mutation, so we further analyzed the mutation type
and function of these 5 genes” (Lines 324-328) in the revised the
manuscript.

First, after the identification of the major Clade 1 strains, we further
analyzed the 152 SNPs found within them, including both the
recombination region mutations and point mutations. All the
mutations were annotated according to their genome positions, from
which 5 genes were identified with more than one mutation, so we
further analyzed the mutation type and function of these 5 genes,
which turned out that they were associated with virulence and
multidrug efflux system.

Second, most of the genes showed negative Tajima’s D values
(Additional file 6: Figure S4), especially the 5 genes mentioned here
which either suggest negative selection or reflect a recent population
expansion. In this case, a recent population expansion was more
likely to happen, thus it might be hard to infer whether they were

under positive or negative selection pressure.

**Point 25:**

**Line289: replace “clades” with “isolates”**

**Response 25:**

Thanks very much for the reviewer's comments, the manuscript had
been revised as suggested.

“Antimicrobial resistance gene content and plasmid carriage of the
outbreak isolates” (Lines 338-339)

**Point 26:**

**Line 304: “highly correlated” is not clear, explain better.**

***bla*_{KPC-2} is only in this type of plasmids. If yes, supply
references**

**Response 26:**

Sorry for our inappropriate description, *bla*_{KPC-2} was not only in this
type of plasmids and we have revised this description in the
resubmitted manuscript.

“All ST11 strains carried the ColRNAI and IncFII plasmid replicons,
which appears to be associated with the presence of several
resistance genes” (Lines 355-357)

**Point 27:**

**Line 311: replace “plasmid genomes” with “plasmid sequence”**

**Response 27:**

Thanks very much for the reviewer's comments, the manuscript had
been revised as suggested. “The plasmid sequences were further
explored using long-read sequencing assemblies.” (Lines 376-377)

**Point 28:**

**Line 328: Add “probably” first of “indicate that....”**

**Response 28:**

Thanks very much for the reviewer's comments, the manuscript had
been revised as suggested. “which probably indicate that strain
xz163 had evolved from a distant parent xz061 containing 3
plasmids.” (Lines 393-395)

**Point 29:**

**Line 347: remove “the” at the beginning of the line**

**Response 29:**

Thanks very much for the reviewer's comments, the manuscript had
been revised as suggested. “which is consistent with previous
results.” (Lines 413-414)

**Point 30:**

**Line 356-359: this phrase is not clear. Rewrite.**

**Response 30:**

Thanks very much for the reviewer's comments, we have revised the
description.

“Based on the spatio-temporal analysis, it was found that the ST11
strains were not the most abundant MLST types in the beginning
during our collection period, the number of ST11 strains had
increased gradually and eventually it became the most common
MLST type.” (Lines 423-426)

**Point 31:**

**Line 364: the term “obtained” is not appropriate**

**Response 31:**

Thanks very much for the reviewer's comments, we have replaced
“obtained” with “acquired”. (Line 431)

**Point 32:**

**Line 381-383: this concept is trivial. Rewrite or remove it**

**Response 32:**

Thanks very much for the reviewer's comments, we have removed
this description as suggested.

**Point 33:**

**Line 389: replace with “plasmids in hospital strains”**

**Response 33:**

Thanks very much for the reviewer's comments, the manuscript had
been revised as suggested. “plasmids in hospital strains.” (Line 454)

**Point 34:**

**Figure 2C: indicate the meaning of dotted lines in the caption**

**Response 34:**

Sorry for causing your confusion, the dotted lines in Figure 2C were
meant to highlight the other strains apart from the ST11 between
which the pairwise number of SNPs were less than 35. For example,
in the Pediatric Intensive Care Unit (PICU), two ST716 strains were
enclosed with a dotted line circle since 8 SNPs were identified
between them indicating that they were also highly clonal.

The Figure 2 legends had been added with detailed description.
“Strains apart from the ST11 between which the pairwise number of
SNPs were less than 35 were enclosed by dotted lines” (Lines
715-717).

**Point 35:**

**Figure 4: I suggest to visualize only genes that are present at**

**least in one genome to simplify the figure. Maybe a completed**
**figure can be put on supplementary.**

**Response 35:**

Thanks very much for the reviewer's comments, all the genes in
Figure 4 were present at least in one genome, we had put it on
supplementary as Additional file 7: Figure S5.

**Reviewer 3:**

The study investigates the population structure and phylogeny of 98
*Klebsiella pneumoniae* isolates obtained from pediatric patients in a
Chinese hospital between February 2018 and May 2019. The authors
reported the clonal spread of KPC-2 producing ST11 *K. pneumoniae*
strains that were clustered into 2 clades. The study described the
plasmid content in the 2 clades complete with resistance genes. The
authors used these findings to explain the higher transmissibility
among clade 1 isolates relative to clade 2 isolates. The study calls for
a wider use of genomic tools to identify outbreaks.

**Minor comments were observed:**

**Point 36:**

**The manuscript needs minor English revision and editing.**

**Response 36:**

Thanks very much for the reviewer's constructive comments, we
have revised the manuscript according to all the reviewers'
comments and the manuscript has also been greatly improved and
reviewed by a native speaker.

**Point 37:**

**-Line 64: Would the authors please explain the importance of**
**carbapenem as last resort agent in the treatment and hence the**
**global concern as a result of carbapenem resistance**
**development?**

**Response 37:**

Thanks very much for the reviewer's comments. Carbapenems are
bactericidal β -lactam antimicrobials with proven efficacy in severe
infections caused by extended spectrum β -lactamase (ESBL)
producing bacteria. They possess broad spectrum antibacterial
activity and have a unique structure that is defined by a carbapenem
coupled to a β -lactam ring which confers protection against most β
lactamases such as metallo- β -lactamase (MBL) as well as extended
spectrum β -lactamases. Consequently, carbapenems are considered
one of the most reliable drugs for treating bacterial infections and the
emergence and spread of resistance to these antibiotics constitute a
major global public health concern. The World Health Organization

recognizes extended-spectrum β -lactam (ESBL)-producing and
carbapenem-resistant *K. pneumoniae* (CRKp) as a critical public
health threat.

**Point 38:**

**-Line 117: was the DNA library prepared at the sequencing**
**company? Please give some details about Illumina library prep.**

**Response 38:**

Thanks very much for the reviewer's comments, the DNA library
was prepared at the sequencing company. We have added detailed
description about Illumina library preparation in the revised
manuscript as suggested

“The DNA library was prepared at the sequencing company. In brief,
genomic DNA was extracted from bacteria colonies using QIAamp
DNA Mini Kit (Qiagen, Hilden, Germany) and quantified using
Qubit fluorometer (Life Technologies, Carlsbad, CA, USA). The
libraries were constructed using the Nextera XT kit (Illumina Ltd.,
San Diego, CA, USA) according to the manufacturer’s
recommendations and sequenced in pair-end mode (2×150 bp)
using the Illumina HiSeq 2500 platform at Novogene Company
(Beijing, China) with a median sequencing depth of 235.6X” (Lines
115-123)

**Point 39:**

**-Line 124-125: Would the authors please give the details of**
**quality check and trimming of the reads?**

**Response 39:**

Thanks very much for the reviewer's comments, we have added
detailed description about quality check and trimming of the reads in
the revised manuscript as suggested

“Sequencing reads were quality filtered using the FastQC v0.11.8
software, adapters and low-quality reads were removed and filtered
out using Trimmomatic with default parameters, the Q30 cut off was
set to 85% and the read length cut off was set to longer than 100bp.”.

(Lines 128-131)

**Point 40:**

**-Would the authors give more details about Fig. 3c as it is not**
**clear how they inferred the substitutions per site per year from**
**the figure.**

**Response 40:**

Thanks very much for the reviewer's comments. we have added
detailed description about how the substitutions per site per year
were inferred in the revised manuscript as suggested

“Briefly, SNPs were predicted using Snippy v4.6.0, the consensus

whole genome alignment was used to infer a maximum likelihood
phylogenetic tree using Raxml-ng v1.0.3. The whole genome
alignment and maximum likelihood phylogenetic tree were
combined to identify and exclude the recombination regions using
ClonalFrameML v1.12. A maximum likelihood phylogenetic tree
based on the non-recombination alignment was constructed using
Raxml-ng v1.0.3. Two major clades were identified, and we assessed
the correlation between root-to-tip distance and the date of isolation
in TempEst v1.5.3. Following the identification of a strong temporal
signal in the Clade 1 strains, we ran BEAST v2.4.7 on the
recombination-filtered chromosomal alignment and used the GTR
substitution model with a coalescent constant population size and a
strict molecular clock rate to infer the substitutions per site per year.”
(Lines 147-184)

**Point 41:**

**-Lines 297-297: Fosfomicin and macrolide susceptibility data**
**are not shown in the table.**

**Response 41:**

Thanks very much for the reviewer's comments, two types of cards
were used for susceptibility testing and some representative
antibiotics were selected in the study. One of the card types is

VITEK 2 AST-N335 test kit, the other one is VITEK 2 AST-GN09
test kit. VITEK 2 AST-N335 test kit include 17 antibiotics (amikacin,
aztreonam, cefepime, cefoperazone/sulbactam, ceftazidime,
ciprofloxacin, colistin, doxycycline, levofloxacin, imipenem
meropenem, minocycline, piperacillin/tazobactam,
ticarcillin/clavulanic acid, tigecycline, tobramycin,
sulfamethoxazole). VITEK 2 AST-GN09 test kit include 20
antibiotics (amikacin, ampicillin, ampicillin/sulbactam, aztreonam,
cefazolin, cefepime, cefotetan, ceftazidime, ceftriaxone, cefuroxime,
ciprofloxacin, gentamicin, imipenem, levofloxacin, meropenem,
nitrofurantoin, piperacillin, piperacillin/tazobactam, tobramycin,
sulfamethoxazole).

We have changed the results description of multiple drug resistance
in the revised manuscript. “Antibiotic susceptibilities of all strains
were determined, and showed multiple drug resistance to
aminoglycosides, beta-lactams and fluoroquinolones”. (Lines
341-344)

**Point 42:**

**-Line 300: Do the authors mean a statistically significant**
**difference?**

**Response 42:**

Thanks very much for the reviewer's comments, we have performed
Pearson's Chi-squared Test for the strains carried the *bla*_{KPC-2} gene
between ST11 and non-ST11 multi-locus sequence types, the
X-squared = 69.997, p-value < 2.2e-16, indicating a statistically
significant difference

We have changed the results description in the revised manuscript.
“observed a significant difference (p<0.001) in the carbapenem
resistance gene content between the ST11 and other STs strains”
(Lines 346-347).

**Point 43:**

**-Lines 304-306: The authors would better comment on the**
**almost complete absence of these genes in the ST11 strains.**

**Response 43:**

Thanks very much for the reviewer's comments. We have added
comments in the revised manuscript as suggested. “The almost
complete absence of these genes in the ST11 strains might be due to
the absence of the plasmid carrying them.” (Lines 353-355)

**Point 44:**

**-Line 306: The authors would better specify that they mean**
**IncFII.pHN7A8_1**

**Response 44:**

Thanks very much for the reviewer's comments. We have revised the
description in the resubmitted manuscript. "In addition, two Clade 1
strains co-carried blaKPC-2 gene and mcr-9 gene which were
located on IncFII.pHN7A8._1 plasmid and IncHI2/IncHI2A plasmid
respectively." (Lines 362-364)

**Point 45:**

**-Line 306-308: "All ST11 strains carried the ColRNAI and**
**IncFII plasmid replicons, which were highly correlated with the**
**presence of blaKPC-2, rmtB, and blaTEM-1B genes" the isolates**
**also carried blaSHV.11_1, blaSHV.155_1 as well as fosA_3**
**coding for Fosfomycin resistance and oqxA_1 and oqxB_1**
**coding for fluoroquinolone resistance.**

**Response 45:**

Thanks very much for the reviewer's comments. We have revised the
description in the resubmitted manuscript.

"All ST11 strains carried the ColRNAI and IncFII plasmid replicons,
which appears to be associated with the presence of several
resistance genes including blaKPC-2, rmtB, blaTEM-1B,
blaSHV.11_1, blaSHV.155_1, fosA_3, oqxA_1 and
oqxB_1(Additional file 7 Figure S5). However, only three genes
(blaKPC-2, rmtB, and blaTEM-1B) among them were found located

on the plasmid contigs indicating that they might be located the two
related plasmids.” (Lines 355-362)

**Point 46:**

**-Line 308: Looks like only 2 clade 1 strains (and not 3)**
**co-carried blaKPC-2 and mcr-9.**

**Response 46:**

Sorry, it was a mistake, we have revised the description in the
resubmitted manuscript. “two Clade 1 strains co-carried blaKPC-2
gene and mcr-9 gene which were located on IncFII.pHN7A8._1
plasmid and IncHI2/IncHI2A plasmid respectively” (Lines 362-364)

**Point 47:**

**-Line 310: Looks like the three mcr-9 positive strains carried the**
**pKPC.CAV1321_1 and not repB_KLEB_VIR plasmid replicon.**

**Response 47:**

Sorry, it was a mistake, we have revised the description in the
resubmitted manuscript. “IncHI2/ IncHI2A and pKPC.CAV1321_1
plasmid replicons were found only in the three mcr-9 positive strains”
(Lines 364-366)

**Point 48:**

**Lines 311-313: the same can be said about ST76 strains and**

**IncX3_1 plasmid replicon and blaNDM.1_1, blaSHV.12_1,**
**fosA_3, oqxA_1 and oqxB_1.**

**Response 48:**

Thanks very much for the reviewer's comments. We have revised the
description in the resubmitted manuscript.

“Seven of the eight ST76 strains carried the IncX3_1 plasmid
replicon which appears to be associated with the presence of several
resistance genes including blaNDM.1_1, blaSHV.12_1, fosA_3,
oqxA_1 and oqxB_1 (Additional file 7 Figure S5). However, only
blaNDM.1_1 gene was found located on the plasmid contigs
indicating that blaNDM.1_1 gene might be located on the IncX3_1
plasmid. These data indicated that ST11 and ST76 strains had
acquired a special plasmid carriage and a corresponding resistance
gene content pattern respectively.” (Lines 367-375)

**Point 49 :**

**Would the authors please comment on Fig 5B at the end of the**
**results? From Fig 3D and Fig 5B, it seems that strain xz163**
**evolved from a distant parent xz061 and acquired a 4th plasmid**
**then further evolved to xz168 and xz181.**

**Response 49:**

Thanks very much for the reviewer's comments, we have added
comments on Fig 5B at the end of the results in the revised
manuscript.

“Three Clade 1 strains (xz163, xz168 and xz181) had acquired a
plasmid (IncHI2/IncHI2A) with co-existence of the *mcr-9*, *mphA*,
and *bla*_{SFO-1} genes. Notably, all of these three strains were isolated
from the Pediatric Intensive Care Unit and clustered closely in both
evolutionary and transmission trees. Thus, it was reasonable to infer
that strain xz163 had evolved from a distant parent xz061 containing
3 plasmids and then acquired a 4th plasmid which co-harbored three
resistance genes and continued to spread to the other two patients in
the same ward.” (Lines 389-397)

**Point 50:**

**Lines 314-316: Please mention Additional file 8 Figure S6 here.**

**Response 50:**

Thanks very much for the reviewer's comments, we have added the
Additional file 9 Figure S7 here.

“Five and four strains were selected from Clade 1 and Clade 2
respectively (Fig. 4A, Additional file 9 Figure S7).” (Lines 377-379)

**Point 51:**

**Lines 371-372: Figures 3D and 3E show the transmission of**

**clade 1 but not clade 2, so unless the authors are referring to**
**other figures here, would they just mention that these results are**
**not shown?**

**Response 51:**

Thanks very much for the reviewer's comments, we also have
conducted the transmission analysis of clade 2, as there were only 4
isolates of clade 2, which were not sufficient to infer the
transmission tree by using the TransPhylo software and the results
were not shown.

We have revised the description in the resubmitted manuscript.
“Transmission analysis results showed that Clade 1 strains had
infected more children and spread to more wards than Clade 2
strains (results not shown).” (Lines 434-436)

October 19, 2022

Dr. Peng Li
Chinese PLA Center for Disease Control and Prevention
DONGDA street 20#
Beijing
China

Re: Spectrum01919-22R1 (Clonal spread of carbapenem-resistant *Klebsiella pneumoniae* ST11 in Chinese pediatric patients)

Dear Dr. Peng Li:

Your paper has been further revised and all the reviewers suggest other modifications and to address completely previous comments

Link Not Available

Sincerely,

Maria De Francesco

Journals Department
Reviewer comments:

Reviewer #1 (Comments for the Author):

The bacterial isolation and AST have been simplified rather than elaborated and explained further. How the bacteria were isolated, at what point identification and antimicrobial susceptibilities were performed are not explained. A review of the electronic medical records should not be explained in the methodology for bacterial isolation and antimicrobial susceptibility. Access to the patients' personal medical files should be explained in the ethical approval as well.

The results of the hand hygiene audit were not provided, instead a detailed explanation of the infection prevention measures was provided. Hand hygiene audit results would indicate the compliance of staff in performing hand hygiene in the wards and

the audits are usually conducted regularly by the infection prevention staff.

Reviewer #2 (Comments for the Author):

Authors improved the quality of the manuscript, adding the experimental and methodological information required by reviewers. Despite this, some formulations and statements remain still inadequate or need to be synthesized. Regarding the style, the final manuscript, in particular some new statements, would benefit from review by a native speaker. A control of the correct use of gene nomenclature (italic, subscript character ect.) is needed.

Reviewer #3 (Comments for the Author):

Would the authors kindly add their response to point 37 to the manuscript?

In the authors' response to point 42, they explained that they carried out Pearson's Chi-squared Test for the strains carrying the blaKPC-2 gene between ST11 and non-ST11 isolates. The authors are kindly requested to add the statistical analysis to the methods section?

Staff Comments:

Preparing Revision Guidelines

Please return the manuscript within 60 days; if you cannot complete the modification within this time period, please contact me. If you do not wish to modify the manuscript and prefer to submit it to another journal, please notify me of your decision immediately so that the manuscript may be formally withdrawn from consideration by Microbiology Spectrum.

**REVIEWER n2**

Authors improved the quality of the manuscript, adding
experimental and methodological information required by reviewers.
Despite this, some formulations and statements remain still
inadequate or need to be synthesized (see “Reviewer New Comment”
in section "Comments on reviewer requests"). Regarding the style,
the final manuscript, in particular some new statements, would
benefit from review by a native speaker.
A control of the correct use of gene nomenclature (italic, subscript
character ect.) is needed.

**COMMENTS ON REVIEWER REQUESTS**

**Point 4:**

**Line 110-115: the phrase should be simplified.**

**Response 4:**

Thanks very much for the reviewer's comments, we have simplified
the phrase. “MICs were interpreted according to the CLSI (2021)
guidelines. The electronic medical records of culture-positive
children were reviewed retrospectively to obtain demographic and
clinical data.” (Lines 110-113)

**Reviewer New Comment:**

Line 108: Cancel the dot after the bracket.

Lines 111-113: Cancel “The electronic medical records of
culture-positive children were reviewed retrospectively to obtain
demographic and clinical data.”. It’s obvious.

**Point 7:**

**Line 124: how reads are quality filtered? Please specify Q30 cut**
**off and read length cut off (eventually)**

**Response 7:**

Thanks very much for the reviewer's comments, we have described
how reads were quality filtered in the revised manuscript.
“Sequencing reads were quality filtered using the FastQC v0.11.8
software, adapters and low-quality reads were removed and filtered
out using Trimmomatic with default parameters.” (Line 128-130)

We also have specified the Q30 cut off and read length cut off in the
revised manuscript. “The Q30 cut off was set to 85% and the read
length cut off was set to longer than 100bp.” (Lines 130-131)

**Reviewer New Comment:**

Lines 130-131: Cancel “The Q30 cut off was set to 85% and the read

length cut off was set to longer than 100bp.”. The first statement is
sufficient.

**Point 11:**

**Line 134 and 138: What SNP pipeline do you use as input for**
**phylogenesis? Is not clear if snippy o roary. Please clarify.**

**Response 11:**

Thanks very much for the reviewer's comments, we have added the
SNP pipeline used as input for phylogenesis in the revised
manuscript.

“Initially, SNPs across the whole genome were predicted using
Snippy v4.6.0, the consensus whole genome alignment was used to
infer a maximum likelihood phylogenetic tree using Raxml-ng
v1.0.3 implementing with 1000 bootstrap replicates, GTR+ was
selected as the best evolutionary model by using Modeltest-ng
v0.1.7” (Lines 147-151)

**Reviewer New Comment:**

Lines 147-151: in order to simplify the text, cancel “across the
whole genome” (line 147) and “GTR+ was selected as the best
evolutionary model by using Modeltest-ng v0.1.7” (line 150-151).

**Point 21:**

**Line 238: how do you choose this threshold? Do you have**
**references? If yes, please add it. It's a very high number of SNPs**
**and this high threshold may be misleading. This is a very**
**IMPORTANT aspect**

**Response 21:**

Thanks very much for the reviewer's comments.

First, we have added description on how the threshold was chosen in
the revised manuscript.

“The distribution of the pair-wise SNPs distances among strains was
visualized as showed in the Additional file 5 Figure S3A, the SNP
cut-off threshold was determined manually by looking at the graph
and finding that there were two distinctive groups of genome pairs,
those with less than 35 SNPs and those with more than 35 SNPs,
which allowed us to safely infer that two genomes could be
considered as part of the same transmission cluster if their distance
in number of SNPs was lower than the 35 SNPs threshold value.”

(Lines161-168)

Second, we also have found a reference in which pairs of genomes
within a distance of 35 SNPs were considered as part of the same
transmission cluster, and we have added it in the revised manuscript.

“Pairs of genomes within a distance of ≤ 35 SNPs (42) were
considered as the same transmission cluster.” (Lines 283-284)

**Reviewer New Comment:**

The reference you mentioned (42) is appropriate but the threshold
indicated for related genome clusters in that paper is 16 (not 35),
which is coherent with other recent multicenter study that proposed
to set a threshold for SNP distance in Kp outbreak at 21 (Davis et al,
2019; see in bibliography of your reference #42). You can propose a
new threshold cut off only if you have strong epidemiological
evidences of correlation (e.g. same patient etc) or a strong evidences
of a longtime evolution that bring to accumulate SNPs in a clone.

I suggest you to add also this last reference (Davis et al, 2019) and to
adjust the threshold value and the results of cluster belonging (with
the new threshold, are all your strains part of the genomic cluster?).

**Point 22:**

**Line 241: how do you calculate that temporal signal was strong?**
**(e.g. Tempest software)**

**Response 22:**

Thanks very much for the reviewer's comments, we have added the

description on how the temporal signal was calculated in the revised
manuscript.

“We used a root to tip regression of sampling dates against genetic
diversity in TempEst v1.5.3, optimizing the best fit for the root to
maximize the determination coefficient R^2 . The slope of the
regression was positive, and the p-value is 0.0006, showing that the
genomic data reflect strong temporal signal.” (Lines 169-173)

**Reviewer New Comment:**

Lines 171-173: in order to simplify the text, cancel “The slope of the
regression was positive, and the p-value is 0.0006, showing that the
genomic data reflect strong temporal signal.”

**Point 24:**

**Line 277: clarify the reason why you mentioned these genes (this**
**gene are under pression in your opinion?). These observations**
**are not well explained.**

**Response 24:**

Thanks very much for the reviewer's comments.

We have changed “In addition, we identified 5 genes with more than
one SNP.” with “In this case, a recent population expansion was
more likely to happen. All the mutations were annotated according to

their genome positions, from which 5 genes were identified with
more than one mutation, so we further analyzed the mutation type
and function of these 5 genes” (Lines 324-328) in the revised the
manuscript.

First, after the identification of the major Clade 1 strains, we further
analyzed the 152 SNPs found within them, including both the
recombination region mutations and point mutations. All the
mutations were annotated according to their genome positions, from
which 5 genes were identified with more than one mutation, so we
further analyzed the mutation type and function of these 5 genes,
which turned out that they were associated with virulence and
multidrug efflux system.

Second, most of the genes showed negative Tajima’s D values
(Additional file 6: Figure S4), especially the 5 genes mentioned here
which either suggest negative selection or reflect a recent population
expansion. In this case, a recent population expansion was more
likely to happen, thus it might be hard to infer whether they were
under positive or negative selection pressure.

**Reviewer New Comment:**

Lines 322-328: in order to simplify the text, substitute only with

“Most of the genes showed negative Tajima’s D values (Additional

file 6: Figure S4), in particular 5 genes with more than one SNP,
suggesting a possible negative selection or a recent population
expansion.”

**Point 26:**

**Line 304: “highly correlated” is not clear, explain better.**

**blaKPC-2 is only in this type of plasmids. If yes, supply**

**references**

**Response 26:**

Sorry for our inappropriate description, *bla*_{KPC-2} was not only in this
type of plasmids and we have revised this description in the
resubmitted manuscript.

“All ST11 strains carried the ColRNAI and IncFII plasmid replicons,
which appears to be associated with the presence of several
resistance genes” (Lines 355-357)

**Reviewer New Comment:**

Lines 359-375: in order to simplify the text, substitute with

“However, only three genes (*bla*KPC-2, *rmtB*, and *bla*TEM-1B)

among them were found located on plasmid contigs. In addition, two

Clade 1 strains co-carried *bla*KPC-2 gene and *mcr-9* gene, which

were located on IncFII pHN7A8.1 plasmid and IncHI2/IncHI2A
plasmid respectively. IncHI2/ IncHI2A and pKPC.CAV1321_1
plasmid replicons were found only in the three mcr-9 positive strains.
Seven of the eight ST76 strains carried the IncX3_1 plasmid
replicon which appears to be associated with the presence of several
resistance genes including blaNDM.1_1, blaSHV.12_1, fosA_3,
oqxA_1 and oqxB_1 (Additional file 7 Figure S5). However, only
blaNDM.1_1 gene was found located on plasmid contigs.

Line363: Please control if it's correct "IncFII pHN7A8.1". Do you
refer to the same plasmid (pHN7A8.1 is a IncFII plasmid?) or two
plasmids are present?

**Point 30:**

**Line 356-359: this phrase is not clear. Rewrite.**

**Response 30:**

Thanks very much for the reviewer's comments, we have revised the
description.

“Based on the spatio-temporal analysis, it was found that the ST11
strains were not the most abundant MLST types in the beginning
during our collection period, the number of ST11 strains had
increased gradually and eventually it became the most common
MLST type.” (Lines 423-426)

**Reviewer New Comment:**

Lines 423-426: Replace with “Based on the spatio-temporal analysis,
it was found that the ST11 was not the most abundant ST type at the
beginning of the collection period, but it increased gradually
becoming the most common ST type in collected strains”

**Point 34:**

**Figure 2C: indicate the meaning of dotted lines in the caption**

**Response 34:**

Sorry for causing your confusion, the dotted lines in Figure 2C were
meant to highlight the other strains apart from the ST11 between
which the pairwise number of SNPs were less than 35. For example,
in the Pediatric Intensive Care Unit (PICU), two ST716 strains were
enclosed with a dotted line circle since 8 SNPs were identified
between them indicating that they were also highly clonal.

The Figure 2 legends had been added with detailed description.

“Strains apart from the ST11 between which the pairwise number of

SNPs were less than 35 were enclosed by dotted lines” (Lines
715-717).

**Reviewer New Comment:**

Lines 715-717: Replace with “Dotted lines indicate strains, other
than ST11, with less than 35 pairwise SNPs”

Response to the reviewers' comments:

Reviewer #1 (Comments for the Author):

Point 1:

The bacterial isolation and AST have been simplified rather than elaborated and explained further. How the bacteria were isolated, at what point identification and antimicrobial susceptibilities were performed are not explained.

Response 1:

Thanks very much for the reviewer's comments, we have elaborated and explained the bacterial isolation and AST as your suggestion.

“The bacteria strains were isolated from clinical specimens, which included sputum, urine, blood, and other samples. Strain identification was performed by matrix-assisted laser desorption/ionization time-of-flight mass spectrometry. In vitro antimicrobial susceptibility testing of isolates was analyzed with a VITEK-2 compact system (bioMérieux, Marcy-l'Étoile, France). Antimicrobial susceptibility testing was interpreted in accordance with the Clinical and Laboratory Standards Institute (CLSI), except for tigecycline and colistin, which were interpreted based on the European Committee for Antimicrobial Susceptibility Testing

(EUCAST) criteria.” (Lines 116-126)

Point 2:

A review of the electronic medical records should not be explained in the methodology for bacterial isolation and antimicrobial susceptibility.

Response 2:

Done as suggested.

Point 3:

Access to the patients' personal medical files should be explained in the ethical approval as well.

Response 3:

Thanks very much for the reviewer's comments, we have explained the access to the patients' personal medical files in the Ethics approval and consent to participate section.

“A guardian of each child patient provided written informed consent for study participation before enrollment. This study was conducted in accordance with the Declaration of Helsinki. The Clinical Research Ethics Committee of the Affiliated Hospital of Xuzhou Medical University approved the study (XYFY2015-JS016-01), as all samples evaluated in this study were initially collected for

diagnosis during patient care and were thereby obtained without increasing the patients' medical costs and suffering.” (Lines 469-476)

Point 4:

The results of the hand hygiene audit were not provided, instead a detailed explanation of the infection prevention measures was provided. Hand hygiene audit results would indicate the compliance of staff in performing hand hygiene in the wards and the audits are usually conducted regularly by the infection prevention staff.

Response 4:

Thanks very much for the reviewer's comments, the hand hygiene compliance rate and correct rate of staff in pediatric ward were 76.9% and 81.64%, respectively.

The hospital had a comprehensive and systematic hand hygiene testing process:

1. Formulate the standard instruction for hand hygiene monitoring of medical staff (document number: CML-SOP-3002);
2. The department of hospital sense co-ordination, supervision and floor departments regularly carry out hand hygiene, and all samples are sent to microbiology laboratory for cultivation and identification

by professionals;

3. Develop environmental hygiene software independently, and trace the results, so as to achieve the purpose of special management and closed-loop monitoring;

4. For medical staff in key departments (such as operating rooms and intensive care units), it is mandatory to monitor hand hygiene at least once a month;

5. In view of the monitoring results of nosocomial risk, the Department of Nosocomial Infection took the lead and conducted targeted hand hygiene monitoring for the corresponding personnel several times.

Hand hygiene examination is divided into surgical hand disinfection effect monitoring and sanitary hand disinfection effect monitoring (after general medical staff wash their hands). The positive rate of surgical hand disinfection effect monitoring in our hospital from 2018 to 2019 was 2.89%, and that of sanitary hand disinfection effect monitoring was 11.19%.

Reviewer #2 (Comments for the Author):

Point 5:

Authors improved the quality of the manuscript, adding the experimental and methodological information required by reviewers. Despite this, some formulations and statements remain still inadequate or need to be synthesized. Regarding the style, the final manuscript, in particular some new statements, would benefit from review by a native speaker.

Response 5:

Thanks very much for the reviewer's all constructive comments, we have carefully revised our manuscript according to the reviewer's comments and improved the manuscript by a native speaker.

Point 6:

A control of the correct use of gene nomenclature (italic, subscript character ect.) is needed.

Response 6:

Thanks very much for the reviewer's comments, we have carefully examined the use of gene nomenclature in our manuscript according to the reviewer's comments.

“*bla*_{KPC-2}, *bla*_{NDM.1_1}, *bla*_{NDM-5}, *bla*_{IMP-4}, *bla*_{TEM-1B}, *bla*_{SHV.11_1}, *bla*_{SHV.12_1},

*bla*_{SHV.155_1}, *bla*_{SFO-1}, *mcr-9*, *mphA*, *rmtB*, *fosA_3*, *oqxA_1* and *oqxB_1*”

Point 7:

Line 108: Cancel the dot after the bracket.

Response 7:

Done as suggested.

Point 8:

Lines 111-113: Cancel “The electronic medical records of culture-positive children were reviewed retrospectively to obtain demographic and clinical data.”. It’s obvious.

Response 8:

Done as suggested.

Point 9:

Lines 130-131: Cancel “The Q30 cut off was set to 85% and the read length cut off was set to longer than 100bp.” The first statement is sufficient.

Response 9:

Done as suggested.

Point 10:

Lines 147-151: in order to simplify the text, cancel “across the whole genome” (line 147) and “GTR+ was selected as the best evolutionary model by using Modeltest-ng v0.1.7” (line 150-151).

Response 10:

Thanks very much for the reviewer's comments, we have revised the two sentences as suggested.

“Initially, SNPs were predicted using Snippy v4.6.0, the consensus whole genome alignment was used to infer a maximum likelihood phylogenetic tree using Raxml-ng v1.0.3 (35) implementing the GTR+ model with 1000 bootstrap replicates.” (Lines 158-161)

Point 11:

The reference you mentioned (42) is appropriate but the threshold indicated for related genome clusters in that paper is 16 (not 35), which is coherent with other recent multicenter study that proposed to set a threshold for SNP distance in Kp outbreak at 21 (Davis et al, 2019; see in bibliography of your reference #42). You can propose a new threshold cut off only if you have strong epidemiological evidences of correlation (e.g. same patient etc) or a strong evidences of a longtime evolution that bring to accumulate SNPs in a clone. I suggest you to add also this last reference (Davis et al, 2019) and to adjust the threshold value and the results of cluster belonging (with

the new threshold, are all your strains part of the genomic cluster?).

Response 11:

Thanks very much for the reviewer's constructive comments.

The pairwise number of SNPs among Clade 1 in our study ranged from 0 to 29, five strains were excluded from Clade 1 with the threshold of 21 SNPs. Strains in the reference (Ferrari et al., 2019) had a time span of 155 days, while our five strains had a time span of 306 days with the first isolated strain and were isolated at the end of the collection period, suggesting a possible longer evolution.

In our analysis (Additional file 5 Figure S3), the strains were divided into two groups, one group had less than 30 SNPs difference among each other, while the other group had more than 43 SNPs. Thus, we had reduced threshold and set the value to 30.

We had added the reference as suggested and revised the description as

“Pairs of genomes within a distance less than 30 SNPs were considered as the same transmission cluster (Fig. 3B, Additional file 5: Figure S3A)” (Lines 294-296)

Point 12:

Lines 171-173: in order to simplify the text, cancel “The slope of the

regression was positive, and the p-value is 0.0006, showing that the genomic data reflect strong temporal signal.

Response 12:

Done as suggested.

Point 13:

Lines 322-328: in order to simplify the text, substitute only with “Most of the genes showed negative Tajima’s D values (Additional file 6: Figure S4), in particular 5 genes with more than one SNP, suggesting a possible negative selection or a recent population expansion.”

Response 13:

Done as suggested.

“Most of the genes showed negative Tajima’s D values (Additional file 6: Figure S4), in particular 5 genes with more than one SNP, suggesting a possible negative selection or a recent population expansion.” (Lines 333-336)

Point 14:

Lines 359-375: in order to simplify the text, substitute with “However, only three genes (*bla*_{KPC-2}, *rmtB*, and *bla*_{TEM-1B}) among them were found located on plasmid contigs. In addition, two Clade

1 strains co-carried *bla*_{KPC-2} gene and *mcr-9* gene, which were located on IncFII pHN7A8.1 plasmid and IncHI2/IncHI2A plasmid respectively. IncHI2/ IncHI2A and pKPC.CAV1321_1 plasmid replicons were found only in the three *mcr-9* positive strains. Seven of the eight ST76 strains carried the IncX3_1 plasmid replicon which appears to be associated with the presence of several resistance genes including *bla*_{NDM.1_1}, *bla*_{SHV.12_1}, *fosA_3*, *oqxA_1* and *oqxB_1* (Additional file 7 Figure S5). However, only *bla*_{NDM.1_1} gene was found located on plasmid contigs.

Response 14:

Done as suggested. (Lines 366-376)

Point 15:

Line363: Please control if it's correct "IncFII pHN7A8.1". Do you refer to the same plasmid (pHN7A8.1 is a IncFII plasmid?) or two plasmids are present?

Response 15:

Sorry for causing your confusion, pHN7A8.1 is a IncFII type plasmid, IncFII pHN7A8.1 was meant to represent the same plasmid, we have revised the sentences as

"IncFII plasmid pHN7A8.1" (Line 369)

Point 16:

Lines 423-426: Replace with “Based on the spatio-temporal analysis, it was found that the ST11 was not the most abundant ST type at the beginning of the collection period, but it increased gradually becoming the most common ST type in collected strains.

Response 16:

Done as suggested. (Lines 423-426)

Point 17:

Lines 715-717: Replace with “Dotted lines indicate strains, other than ST11, with less than 35 pairwise SNPs.

Response 17:

Done as suggested. (Lines 736-737)

Reviewer #3 (Comments for the Author):

Point 18:

Would the authors kindly add their response to point 37 to the manuscript?

Response 18:

Thanks very much for the reviewer's comments, we have added the response to point 37 to the revised manuscript.

“Carbapenems are antimicrobials with proven efficacy in serious infections caused by extended spectrum β -lactamase (*ESBL*) producing bacteria. They possess broad spectrum antibacterial activity which confers protection against most β lactamases such as metallo- β -lactamase (MBL) as well as extended spectrum β -lactamases. Therefore, carbapenems are used as the last resort antibiotics for treating bacterial infections. The World Health Organization listed extended-spectrum β -lactam (*ESBL*)-producing and carbapenem-resistant *K. pneumoniae* (*CRKp*) as a critical public health threat.” (Lines 64-73)

Point 19:

In the authors' response to point 42, they explained that they carried out Pearson's Chi-squared Test for the strains carrying the *blaKPC-2* gene between ST11 and non-ST11 isolates. The authors are kindly requested to add the statistical analysis to the methods section?

Response 19:

Thanks very much for the reviewer's comments, we have added the statistical analysis to the revised methods section.

“Pearson's Chi-squared Test was performed for the strains carried the *bla_{KPC-2}* gene between ST11 and non-ST11 multi-locus sequence

types, the X-squared and p -value were calculated to infer statistically significant difference.” (Lines 201-204)

October 28, 2022

Dr. Peng Li
Chinese PLA Center for Disease Control and Prevention
DONGDA street 20#
Beijing
China

Re: Spectrum01919-22R2 (Clonal spread of carbapenem-resistant *Klebsiella pneumoniae* ST11 in Chinese pediatric patients)

Dear Dr. Peng Li:

I am pleased to inform you that your paper has been accepted for publication

Your manuscript has been accepted, and I am forwarding it to the ASM Journals Department for publication. You will be notified when your proofs are ready to be viewed.

Sincerely,

Maria De Francesco
Editor, Microbiology Spectrum
